# Pedestrian Detection Using Integrated Aggregate Channel Features and Multitask Cascaded Convolutional Neural-Network-Based Face Detectors

**DOI:** 10.3390/s22093568

**Published:** 2022-05-07

**Authors:** Jing Yuan, Panagiotis Barmpoutis, Tania Stathaki

**Affiliations:** 1Department of Electrical and Electronic Engineering, Faculty of Engineering, Imperial College London, London SW7 2AZ, UK; t.stathaki@imperial.ac.uk; 2Department of Computer Science, University College London, London WC1E 6EA, UK; p.barmpoutis@ucl.ac.uk

**Keywords:** pedestrian detection, combination of detectors, aggregate channel features, multitask cascaded convolutional networks

## Abstract

Pedestrian detection is a challenging task, mainly owing to the numerous appearances of human bodies. Modern detectors extract representative features via the deep neural network; however, they usually require a large training set and high-performance GPUs. For these cases, we propose a novel human detection approach that integrates a pretrained face detector based on multitask cascaded convolutional neural networks and a traditional pedestrian detector based on aggregate channel features via a score combination module. The proposed detector is a promising approach that can be used to handle pedestrian detection with limited datasets and computational resources. The proposed detector is investigated comprehensively in terms of parameter choices to optimize its performance. The robustness of the proposed detector in terms of the training set, test set, and threshold is observed via tests and cross dataset validations on various pedestrian datasets, including the INRIA, part of the ETHZ, and the Caltech and Citypersons datasets. Experiments have proved that this integrated detector yields a significant increase in recall and a decrease in the log average miss rate compared with sole use of the traditional pedestrian detector. At the same time, the proposed method achieves a comparable performance to FRCNN on the INRIA test set compared with sole use of the Aggregated Channel Features detector.

## 1. Introduction

Pedestrian detection, a fundamental task in computer vision, can assist in self-driving, the monitoring of crowded environments, and other activities by automatically detecting and localizing human bodies from images. This type of task is challenging, mainly because human bodies are depicted in numerous ways, hindering the overall description of their features. Firstly, human bodies distinguish themselves from each other in appearance, for example, the color of the skin, clothes, hairstyle, body structure, and pose. Secondly, human bodies often suffer from occlusion in crowded environments, such as train stations, shopping malls, and others. Occlusion reduces the area of a body exposed to smaller irregular-shaped areas. Lastly, images are sensitive to the image acquisition setup, such as the varying illumination conditions, view angles, and resolution. For instance, poor illumination or overexposure result in low-contrast images, which blur the human bodies. Furthermore, the human body shape appears different from various viewing angles, for example, from frontal and profile views. Finally, low-resolution human bodies cannot be easily identified, even by the human eye. To handle the diversity of human bodies, it is imperative to describe their appearances effectively using a robust and rich feature set that allows human bodies to be discriminated effectively from the background.

Feature extractors based on deep learning have drawn extensive attention in recent years due to their outstanding performance in extracting high-level semantic information and its large number of features, which has improved the detection performance dramatically. Those methods require training on extremely large datasets to learn representative and robust features; otherwise, the extracted features may be not generalized enough. Because of this, advanced deep learning methods often require massive volumes of annotated data and computational resources such as high-performance computers with GPU facilities. Considering this dilemma, many researchers have focused on the improvement of widely used traditional handcrafted features or the combination of Convolutional Neural Networks (CNNs) with manually extracted features for various computer vision tasks, including pedestrian detection [1,2,3,4,5,6].

One of the earliest popular families of handcrafted features for describing object silhouettes is the histogram of edge orientations, which was initially used for hand gesture recognition [7]. These features are signals that represent occurrences of gradient orientations in localized portions of an image. Another popular contour-based model is the so-called Shape Context [8], designed for measuring shape similarity for the purpose of shape matching. It is defined as a histogram of the relative coordinates of a reference point to a predefined set of neighboring points. This feature is suitable for matching objects whose contours (edges) are corrupted by weak noise or are noiseless and are placed on simple or ideally uniform backgrounds. Considering the complexity of real-life crowded environments, this feature is not recommended for use in human detection. In contrast to those two features, Haar-like features have been used for human detection [9,10] since the early stages. An image is filtered by several Haar wavelets with different predesigned patterns to extract edge features, line features, and center-surround contrast features. The Haar-like feature method is more suitable for objects with simple structures, for example, facial features, which have relatively simple structures. Otherwise, highly textured structures, like grass and trees, which challenge the sufficient representation of predesigned limited Haar wavelets can cause false positives [10]. Another of the most well-known features is the Scale-Invariant Feature Transform (SIFT) model, which consists of the position, scale, and orientations at selected key points, which are “interesting” points of the image signal, for example, contour corners [11]. An advanced version of SIFT is the Principal Components Analysis SIFT (PCA-SIFT) which uses PCA to represent the normalized gradient key point patches, proposed by Ke and Sukthakar in [12]. The authors of this work demonstrated that this method considerably improves both the accuracy and speed compared with the standard SIFT.

In [13], Dalal and Triggs compared the PCA-SIFT, wavelets, and shape context features with the Histogram of Oriented Gradients (HOG) for pedestrian detection and demonstrated that the HOG features greatly outperform the other features on both the MIT and INRIA pedestrian datasets. Owing to [13], HOG has become one of the most widely used features to extract silhouette information for human and other types of object detection. The HOG calculates occurrences (histograms) of gradient orientation in localized portions of an image. Gradient magnitudes serve as weights (votes) to strengthen or weaken the contributions of orientations. This technique differs from the methods mentioned above in that it is computed on a dense grid of uniformly spaced cells and uses overlapping local contrast normalization for improved accuracy. The HOG feature is the concatenation of histograms in overlapping blocks.

To further improve the detection results produced by HOG, some research has focused on enriching the HOG feature sets by combining them with additional cues. The celebrated Aggregated Channel Features (ACF) [14], variants of the Integral Channel Features (ICF) [15], were proposed. The ACF detector begins with the computation of a color channel, such as RGB, LUV or HSV, the magnitude channel, and the HOG channel. These channels are aggregated and vectorized into an enhanced feature vector before being sent to the classifier. As mentioned previously, the ACF detector is characterized by taking both the color feature and the silhouette feature into consideration. Such rich features outperform the HOG feature in the detection of human bodies with various appearances. However, the ACF detector still misses some partly occluded human bodies due to the loss of feature-related information.

Compared to the ACF detector, the subsequent Deformable Part Models (DPM) detector [16] aims to create more sophisticated models based on geometric deformations of a canonical configuration of object parts. It employs a star-structured graph model that can represent body parts and their geometric relations and consists of multiple part filters. Each filter mainly adopts the HOG feature and matches one part or the whole shape of the human body. The transformed responses of filters are output as the combined score of the root location, which defines a detection window approximately covering the entire object.

Inspired by the DPM detector, we adopt the approach of combining multiple detectors. The main advantage of this idea is that even an occluded human body can have a relatively high confidence score as long as the remaining visible body parts are detected and classified. Detecting human bodies by parts is, however, not a novel idea [9,16]. A novel contribution of our work is to incorporate the face detector rather than the commonly used body part detectors. This is because facial features, if available, are more discriminative than other body parts, such as the limbs and the head. Furthermore, the face detector can assist in the detection of human bodies when only the face is visible, a scenario which is quite realistic in crowded environments. In this work, we opt to combine the ACF detector with a face detector. The reasons behind selecting the ACF instead of the DPM are twofold. Firstly, we anticipate, based on preliminary investigations, that the joint use of the DPM and the face detector will yield a higher false positive rate. Furthermore, we wish to take advantage of the superior computational speed of ACF in comparison with that of DPM.

We aim to improve the performance of pedestrian detectors with only small datasets and limited computational resources available through the guidance of faces. To achieve this goal, we adaptively fuse the Multi-Task Cascaded convolutional Neural Networks (MTCNN) detector proposed in [17] with the ACF detector via the proposed score combination module. This module matches the most appropriate face for each human body and gives a comprehensive score for the two. The proposed integrated detector not only has enriched features but is also capable of identifying visible faces. Additionally, the pretrained MTCNN can be directly applied to small datasets without retraining. Therefore, the proposed detector benefits from deep models while avoiding the problems introduced by training on small datasets at the same time. The integrated detector is constructed by scaling and accumulating the face scores to the body scores to yield a final overall score which stands for an overall decision that arises from both detectors. The scaled face scores are higher when faces are located closer to the position at which a face is most likely to appear. Finally, the integrated detector outputs body bounding boxes and their scores taking account of the color, silhouette, and effective face features. In comparison to the sole use of the pretrained ACF detector provided by Piotr’s toolbox, the proposed detector successfully increases the precision level from 92.19% to 93.21% with the average miss rate decreasing from 16.85% to 14.29% on the INRIA pedestrian test dataset, even lower than that of YOLOv3 [18] (14.75%) and comparable to that of FRCNN [19] (14%). It also performs better on the ETHZ, Caltech, and Citypersons datasets.

Our main contributions are summarized as follows:A novel pedestrian detector integrating the multitask cascaded CNN and ACF is proposed;Improved detection performance for significantly occluded pedestrians and beyond is achieved;Robustness of the proposed detector in terms of datasets and beyond is achieved.

The paper is organized as follows: Section 2 presents the methodology used to construct the integrated detector, including the technical procedures involved and detailed explanations regarding the function and design principles of each module in the proposed detector. In Section 3, we investigate the choices of parameters and test the proposed detector on various datasets. The discussion is presented in Section 4.

## 2. Materials and Methods

### 2.1. The Proposed Detector

Traditionally, the ACF human body detector takes the features of a sliding window $\left(b_{x},b_{y},b_{w},b_{h}\right)$ as the input. The two-dimensional vector $\left(b_{x},b_{y}\right)$ contains the *x* and *y* Cartesian coordinates of the top left corner of the sliding window, which has a width of $b_{w}$ and a height of $b_{h}$, as denoted in Figure 1. Our integrated detector differs from the ACF human body detector, mainly by introducing a face detector module and score combination, the output of which is then fed into the ACF detector together with the features, as shown in Figure 2.

The proposed integrated detector starts with feeding an image into a face detector which outputs N predicted face bounding boxes $\left(f_{x}^{n},f_{y}^{n},f_{w}^{n},f_{h}^{n}\right)$ with centers located at $\left(f_{x}^{n}+f_{w}^{n}/2,f_{y}^{n}+f_{h}^{n}/2\right)$ and corresponding face scores of $s^{n}$, n=1,2,…N. The vector $\left(f_{x}^{n},f_{y}^{n}\right)$ stands for the x and y coordinates of the top left corner of the n-th face bounding box with a width of $f_{w}^{n}$ and a height of $f_{h}^{n}$, as shown in Figure 1. Considering that the MTCNN face detector outputs an array of face probabilities as face scores, these face scores are transformed to weights according to
(1)$$s_{f}^{n}=\frac{1}{2}\ln\left(\frac{s^{n}}{1-s^{n}}\right)$$
to keep consistent with the ACF scores as the first step of the score combination.

Secondly, face bounding boxes are filtered following two rules. One rule is to eliminate face bounding boxes that are beyond the enlarged sliding window area shown as the dotted box in Figure 1. The height and the width of the enlarged sliding window are increased by 2×offset compared with those of the original sliding window. The other rule is to eliminate relatively large face bounding boxes determined by the sliding window and two predesigned parameters, namely $r_{w}$ and $r_{h}$. The remaining M face bounding boxes meet the following conditions:(2)$$f_{x}^{m}\geq b_{x}-offset$$
(3)$$f_{y}^{m}\geq b_{y}-offset$$
(4)$$f_{x}^{m}+f_{w}^{m}\leq b_{x}+b_{w}+offset$$
(5)$$f_{y}^{m}+f_{h}^{m}\leq b_{y}+b_{h}+offset$$
(6)$$b_{w}/f_{w}^{m}\geq r_{w}$$
(7)$$b_{h}/f_{h}^{m}\geq r_{h}$$

In the last step of the score combination, the scores ($s_{f}^{m},\;m=1,2,\ldots M$) of the remaining face bounding boxes are scaled to $s_{sc}^{m}$ according to
(8)$$s_{sc}^{m}=f_{ss}\cdot s_{f}^{m}\cdot \left(1-\frac{d^{m}}{d_{0}}\right)$$
where $f_{ss}$ is the face score scale, and $d^{m}$ is the Euclidian distance from the center of the m-th face bounding box to the anchor, which is located at $\left(b_{x}+f_{wr}\cdot b_{w},b_{y}+f_{hr}\cdot b_{h}\right)$. The parameter $d_{0}$ is $d_{0}=d_{hr}\;\cdot b_{h}$, where $d_{hr}$ is the ratio of the Euclidian distance of the zero-scaled score to $b_{h}$. Furthermore, $f_{ss},f_{wr},f_{hr}$, and $d_{hr}$ are predesigned parameters, which are illustrated in Figure 1. The maximum scaled score, if it exists, is assigned to the initial overall score $s_{all}=\max\left(\;s_{sc}^{m}\right)$; otherwise, the initial overall score is assigned as 0. The initial overall score is fed into the ACF detector module to produce the final overall score, which is sent to the threshold module. In this module, the final overall score is considered in two cases according to its initial score. The first case is that, if the initial overall score is assigned by the scaled face score, the sliding window with $s_{all}>s_{thr}$ will be the output as a nominal body bounding box. Otherwise, the sliding window with $s_{all}>-1$ will be the output. Note that −1 is the default threshold of the ACF detector, and $s_{thr}$ is a predesigned parameter that should be larger than −1.

### 2.2. Modules of the Proposed Integrated Detector

In this part, the functionality and design principles of each module shown in Figure 2 are explained in detail.

#### 2.2.1. Sliding Window

This module (Figure 3) outputs the location, size, and ACF features of a cropped area, namely the sliding window, of the image. As explained previously, the location and size are expressed as $\left(b_{x},b_{y},b_{w},b_{h}\right)$ on the original scale. This means that if the sliding window is in a subsampled layer in a feature pyramid, its size and location must be expanded according to the specific subsampling rate. The ACF features consist of a feature vector, which is the concatenation of the LUV color channel, the gradient magnitude channel, and the HOG channel.

#### 2.2.2. Face Detector

This module (Figure 4) takes an image as the input and outputs the detected face bounding boxes jointly with the face scores. Face detection plays a significant role in the overall integrated detection, because the larger the face score, the more likely it is for the corresponding human body to be detected. To correctly identify human bodies, the face detector should have high precision and yield a small number of false positives.

We tested the pretrained Viola Jones (VJ) face detector [20], the fast face detector [21], and the MTCNN detector [22] on the INRIA pedestrian test dataset [13]. According to the results shown in the left column of Figure 5, the VJ detector tends to miss faces that are not presented in the frontal view and those that are occluded. False positives appear when the background is relatively complicated or when there are structures that resemble human faces, such as wheels. The fast face detector was designed based on the work of [23]. This new modified version of the detector extends the ACF used in [23] by adding an integral image channel, in which every pixel is the summation of all of the pixels above and to the left of it. As shown in the middle column of Figure 5, some multiview faces are identified at the expense of a dramatically increasing number of false positives. Compared with these two face detectors, the MTCNN detector gives more accurate detection results with lower false positives rates. As shown in the right column of Figure 5, no false positives occur in these four sample images, and only faces that are largely occluded or presented from mainly the back view are missed, which is expected. This level of performance is due to the collaboration of three convolutional neural networks: the Propose-Network (P-Net), Refine-Network (R-Net), and Output-Network (O-Net). An image is first fed into the fully convolutional neural network P-Net to quickly yield a large number of candidate detections, which are subsequently refined by the R-Net by further correction of the regression vector of the face candidate frame and nonmaximum suppression. The final face regression boxes and facial landmarks (contour key points) are output after correcting and filtering the detections produced earlier with the landmarks, corrections, and probabilities output by the O-Net. The MTCNN detector finally outputs the adjusted face bounding boxes, the facial landmarks, and face scores in the range [0, 1]. Note that more faces are detected by the MTCNN detector without the use of the O-Net, as shown in Figure 6. However, as expected, this structure also results in more false faces, as shown in Figure 7. To achieve the best and most robust face detection, the overall MTCNN detector, therefore, was chosen as our integrated detector.

#### 2.2.3. Score Combination

The score combination consists of three modules, namely the face score transformation module, the face bounding box filtering module, and the score scaling and assigning module.

#### 2.2.4. Face Score Transformation

This module takes the face scores generated by the MTCNN detector as inputs and outputs the corresponding weights, which are the transformed face scores according to Equation (1). This is to remain consistent with the weights produced by the cascading decision trees in the ACF detector. After the transformation, the face detector can be regarded as a single decision tree. The transformed face scores are used in the score scaling and assigning module.

#### 2.2.5. Face Bounding Box Filtering

This filtering module (Figure 8) takes $\left(f_{x}^{n},f_{y}^{n},f_{w}^{n},f_{h}^{n}\right)$ and $\left(b_{x},b_{y},b_{w},b_{h}\right)$, i.e., the sizes and locations of the face bounding boxes and the sliding windows respectively as inputs and decides which face bounding boxes should be sent to the score scaling and assigning module by checking whether they are potential faces of this sliding window according to the two rules explained below.

The first rule is that the true face must be within the potential body bounding box, so the potential faces should be inside the sliding window. There are also face bounding boxes that are only partly inside the sliding box, for example, face bounding box D in Figure 1. To leave some flexibility for such bounding boxes, the four window edges are enlarged by a predesigned offset, shown as the dotted box in Figure 1. This rule is mathematically expressed as (2)–(5). Figure 9 shows an example from the INRIA pedestrian dataset. The overall scores of the ACF detector (a) and the integrated detector (b) are both 57.6. This is because the face (d) should have contributed to the overall score but is filtered out, as it is located at the edge of the sliding window (b). By setting the offset as 5 pixels, the true positive (c) has a higher score of 62.55, as the face score is successfully included.

The second rule is that the face bounding box should not be too large compared with the sliding window. According to the ground truth of the INRIA pedestrian test dataset, only the bounding boxes that contain most parts of a body where the person’s height is larger than 100 pixels are labelled as true positives. Therefore, only the largest bounding boxes shown in Figure 10a–c are true positives, while other smaller bounding boxes are false positives, even though some of them do contain human body parts, such as those shown in Figure 10a. This phenomenon is exacerbated by the incorporation of the face detector in the system, as shown in Figure 10b. This is because the new face scores are large enough to alter the detection results of the ACF detector by introducing false positives when the sliding window is too small to contain sufficient features. To eliminate such false positives, the second rule is adopted by setting the minimum width ratio $r_{w}$ and height ratio $r_{h}$. The width ratio is the ratio of the width of the sliding window to that of the face bounding box, and the height ratio is the ratio of the height of the sliding window to that of the face bounding box. Face bounding boxes with any ratio larger than $r_{w}$ or $r_{h}$ are filtered out, as expressed in (6) and (7). For example, setting $r_{w}=3$ and $r_{h}=8$ eliminates 1 false positive in Figure 10c in comparison with Figure 10b.

#### 2.2.6. Score Scaling and Assigning

This last module (Figure 11) of the score combination takes the filtered face scores $s_{f}^{m}$ and the locations and sizes $\left(f_{x}^{m},f_{y}^{m},f_{w}^{m},f_{h}^{m}\right)\;\mathrm{as}\;\mathrm{inputs}\;$. They are used to compute the scaled score, which is assigned to the initial overall score and fed into the subsequent ACF detection module.

The basic rule of score scaling is that the nearer the face bounding box is to the anchor of a sliding window, the higher the initial overall score is. The anchor is located at the most likely position that a face of an upright human body will appear at. According to this rule, the scaled score is computed by (8), as illustrated in Figure 12 below. The highest scaled score appears at $d^{m}=0$, which means that the face bounding box is located exactly at the anchor, and the sliding window is temporarily considered to be the most likely to contain a human body. The scaled score decreases as the face bounding box moves away from the anchor and reaches 0 when $d^{m}=d_{0}$. The parameter $d_{0}$ is the zero-scaled score distance, as marked in Figure 1. $d_{0}$ should be linearly related to the size of the sliding window to adapt to the changes in the size of the area in which the faces are likely to appear, as introduced by the varying sliding window sizes, so we set $d_{0}=d_{hr}\cdot b_{h}$. The scaled score becomes negative when $d^{m}>d_{0}$ and reaches its minimum value at the maximum value of $d^{m}$ before the face bounding box is filtered out. Such negative scores can help to eliminate false human body bounding boxes. According to (8), the scaled scores of the five face bounding boxes shown in Figure 1 are ranked as A>B>C=0>E>D.

To increase the weight of the MTCNN detector, the predesigned face score scale $f_{ss}$ is introduced in (8), so that the scaled score $s_{sc}^{m}$ of the MTCNN detector is equivalent to the sum score of $f_{ss}$ decision trees. The larger $f_{ss}$ is, the greater the face detector’s influence on the result of the integrated human body detector is. In Figure 13, an example is illustrated, where the human body is missed by the ACF detector (left) but is detected by the integrated detector with $f_{ss}=1$ (right). In another example illustrated in the top sequence of Figure 14, multiple human bodies are missed, although their faces are correctly detected. This is because the face score is overly small compared with the summed score of up to 2048 decision trees in the ACF body detector. As a result, even with the face score included, the overall score of a bounding box associated with a missed body is still too small to reach the threshold. However, we observe in Figure 15 that when $f_{ss}$ is increased to 5, 8, or 10, the previously missed human body is now detected. This means that as the weight of the MTCNN detector increases, the accuracy of a body bounding box increases.

After score scaling, the maximum scaled score $\max\left(s_{sc}^{m}\right)$ is assigned to the overall score of the sliding window for initialization. Sliding windows with positive initial scores are more likely to survive the ACF detector, while those with negative initial scores are more likely to be eliminated.

#### 2.2.7. ACF Detector

The ACF detector (Figure 16) that is employed to detect the human body in the proposed framework is mainly based on the pretrained cascading decision trees [24]. It takes the aggregated color channel, the gradient magnitude channel, and the HOG channel of the sliding window as the input feature vector and outputs the final overall score and the location of the sliding window as the nominal human body bounding box. Note that the final overall score is obtained through the addition of the scaled face score, considered the initial overall score, and the body score. One modification is that the initial overall score of the ACF detector is set as $\max\left(s_{sc}^{m}\right)$, as mentioned in the score scaling and assigning module.

The other modification is that the face threshold $s_{thr}$ is set to filter out the additional false positive sliding windows that contain faces. To show some examples of such false positives and explain their appearances, we compared the ACF detector and the integrated detector with $f_{ss}=8$ and $s_{thr}=-1$ on the INRIA pedestrian test dataset. The number of true positives rose from 543 to 550; however, the number of false positives also rose from 328 to 359. Some samples of additional false positives are shown in the top set of images of Figure 17. The human bodies in these samples have already been correctly marked by the bounding boxes with high body scores, as shown in the bottom set of images in Figure 17, and the bounding boxes may still have relatively high body scores if they are misplaced by only a short distance. After including the scaled face scores, these misplaced boxes are easily identified as false positives. This also explains why each false positive shown in Figure 17 contains a human face. Though the final overall scores of such false positives are higher than the default threshold, they are much lower than those of the true positives, as shown in Figure 17. Considering this phenomenon, a face threshold $s_{thr}$, higher than the default threshold and lower than the final overall scores of the true positives, is set to filter out false positives containing faces. This face threshold must be higher than the default threshold, because the final overall scores are increased by the scaled face scores, whereas the default threshold is designed without considering face scores. The face threshold $s_{thr}$ helps with the elimination of false positives introduced by the incorporation of face scores and the implementation of $f_{ss}$. Note that the default threshold is used if the sliding window does not contain any faces.

The procedure of the proposed integrated detector is summarized in Algorithm 1.
**Algorithm 1****:** Procedure used by the integrated detector**Input**A sliding window (bx,by,bw,bh), face bounding boxes (fxn,fyn,fwn,fhn), face scores sn, n=1,2,…N.**Settings**sall=0, m=0, set offset, rw,  rh, fss, dhr, fwr, fhr and sthr.**Output**Qualified (bx,by,bw,bh), sall.Step 1For each n:If Equations (2)–(7) are fulfilled:
               m=m+1         sscm=fss⋅12ln(sn1−sn)⋅(1−dndhr ⋅bh)             sall=max{ sscm}, m=1,2,…M
Step 2Send sall to the cascading decision trees.Step 3If sall=max{sscm} is implemented:  If sall>sthr:    Output (bx,by,bw,bh);else if sall>sdefault (default threshold):   Output (bx,by,bw,bh).

## 3. Experiments Analysis and Results

In this section, we first investigate the influences of eight predesigned parameters on the performance of the proposed detector to obtain the optimal parameters. Afterwards, the tuned detector is compared with state-of-the-art methods, and its robustness in is evaluated on various datasets.

In the following experiments, the INRIA pedestrian dataset, Caltech pedestrian dataset, Citypersons dataset, and the ETHZ dataset were used.

INRIA pedestrian dataset [13]: The test dataset contains 288 positive color images with 589 labeled human bodies. These images were shot at around eye-level. Most of these human bodies have an upright orientation with some extent of occlusion.Caltech pedestrian dataset [25]: This dataset consists of approximately 250,000 frames, 640 × 480 in size, and a total of 350,000 annotated bounding boxes. The standard test set with 4024 images and corresponding new annotations [26] were used in subsequent experiments. Each image contains about 1.4 persons.Citypersons dataset [27]: The validation set contains 500 high-resolution images, 1024 × 2048 in size, and a total of 3938 persons. Each validation image contains about 7.9 persons.ETHZ dataset [28]: This dataset is a collection of 8 video sequences from busy inner-city locations with annotated human bodies. We assessed two representative sequences from this dataset, namely the BAHNHOF sequence and the Sunny Day sequence. As the pretrained ACF detector cannot classify human bodies with very small sizes, ground truths with widths and heights smaller than 32 and 80 pixels, respectively, were filtered out from the image sequences. After this, the BAHNHOF sequence had 999 images with 3341 ground truths and the Sunny Day sequence had 354 images with 1560 ground truths.

For Caltech and Citypersons, pedestrians were allocated to the reasonable subset, heavily occluded subset, and all subset. The reasonable subset is a collection of pedestrians with heights greater than 50 pixels and visibility levels greater than 0.65. For the heavily occluded subset, the visibility lies in the range [0.2, 0.65]. The all subset consists of pedestrians with heights greater than 20 pixels and a visibility level greater than 0.2.

The MTCNN detector utilized in the experiments is based on the convolutional neural network, which not only detects human faces but also locates facial landmarks. It is available online and is well-trained, and therefore, it was directly applied to our detector. Note that, facial landmark locations were discarded, as only the face bounding boxes and probabilities were used in our detector.

The ACF detector is available in the Piotr’s MATLAB toolbox version 3.40. It is an Adaboost classifier based on cascading binary decision trees. The ACF detector was pretrained on both the INRIA and Caltech pedestrian datasets, respectively.

### 3.1. Parameter Design

The integrated detector has eight predesigned parameters, namely offset, $r_{w}$, $r_{h}$ in the face bounding boxes filtering module, $f_{wr}$, $f_{hr}$, $d_{hr}$, $f_{ss}$ in the score scaling and assigning module, and $s_{thr}$ in the ACF detector. To fully exploit the power of the integrated detector, we investigated the influences of these parameters on the detection results via control variates in the following experiments. The MTCNN face detector available at [22] and the pretrained ACF detector provided by [24] were utilized, and the pre-designed parameters were initially set as offset=0, $r_{w}=3$, $r_{h}=7$, $f_{wr}=1/2$, $f_{hr}=1/8$, $d_{hr}=1/4$, $f_{ss}=8$, $s_{thr}=25$. The integrated detector was tested on the INRIA pedestrian test dataset, which has 589 ground truths. The results were evaluated quantitively by calculating the recall and the log-average miss rate. The recall was calculated as the number of true positives divided by the number of groundtruths. The log-average miss rate refers to the average miss rate over the False Positives Per Image (FPPI) in the range $\left[10^{-2},10^{0}\right],$ which can be calculated automatically using Piotr’s MATLAB toolbox. The miss rate is defined as (1−Recall).

Table 1, Table 2, Table 3, Table 4, Table 5 and Table 6 list the detection results from the integrated detector for various choices of parameters. Their corresponding Receiver Operating Characteristic (ROC) curves are shown in Figure 18. Considering that $s_{thr}$ and $f_{ss}$ are correlated parameters because the face threshold should adapt to the final overall score, their 3D histograms are drawn in Figure 19, instead of using 2D curves and tables. Note that some abbreviations are used in these figures and tables, namely TP (the number of True Positives), FP (the number of False Positives), R (Recall), and AMR (the Average Miss Rate).

According to Table 1 and Figure 18a, the same recall and 3 more false positives are produced when the offset increases from 0 to 6. This shows that the contribution of the offset is not obvious in our setup. A value of offset=0 is recommended when one wishes to minimize false positives. Table 2 and Table 3 and Figure 18b,c show that $r_{w}=1–4$ and $r_{h}=7,\;8$ can produce the maximum number of true positives. Increasing $r_{w}$ and $r_{h}$ leads to less false positives, because relatively small sliding windows are filtered out. However, some true positives are also eliminated when $r_{w}$ and $r_{h}$ are too large. To maintain as many true positives as possible, $r_{w}=1$ and $r_{h}=7$ are recommended. As for the score scaling and assigning, the anchor best locates at the middle of the width ($f_{wr}=0.5$) and 1/8th of the height ($f_{hr}=1/8$), as presented in Table 4 and Table 5 and Figure 18d,e. This location is in line with the face positions of the most upright adult human bodies. Table 6 and Figure 18f show that the best $d_{hr}$ is 1/4. As shown in Figure 19a–c, $s_{thr}$ is inversely proportional to the number of true positives, false positives, and recall, whereas $f_{ss}$ is directly proportional to them. This is because a higher threshold brings in fewer bounding boxes, but more are obtained when the final overall scores are augmented by $f_{ss}$. To strike a balance between these two parameters, parameters of $s_{thr}=25$, $f_{ss}=8$ are suggested, which produces the smallest average miss rate, as shown in Figure 18d.

According to these experimental results, the predesigned parameters can be allocated into four types according to their functions. First, $r_{w},r_{h}$, and $s_{thr}$ can be increased to eliminate false positives but with the sacrifice of some true positives. In contrast, $f_{ss}$ can be increased to bring in both additional true and false positives. Thirdly, choosing appropriate $f_{wr}$, $f_{hr}$, and $d_{hr}$ values can increase the true positives and decrease the false positives at the same time. Finally, the offset has a negligible influence on the detection results.

### 3.2. Evaluation

Following an extensive experimental validation study, we can claim that, for the dataset considered, the integrated detector is finely tuned and produces the best detection results when we choose $offset=0,r_{w}=3,r_{h}=7,\;f_{wr}=0.5,\;f_{hr}=0.125,\;d_{hr}=0.25,\;f_{ss}=8$, $s_{thr}=25$. Furthermore, for the dataset considered, as shown in Figure 20, the integrated detector produces a better performance than the traditional ACF detector with the average miss rate decreasing from 16.85% to 14.29%. By fusing the MTCNN face detector with the body detector, the number of true positives increases from 543 to 549, while the number of false positives decreases from 328 to 319, as shown in Table 7. Some image samples that depict the increased true positives and the eliminated false positives are shown in Figure 21 and Figure 22, respectively.

#### 3.2.1. Comparison with the State-of-the-Art Detectors

We also compared the proposed detector with state-of-the-art detectors, including the handcrafted feature based detectors HOG + SVM [13], DPM [16], ACF [14], and ACF + HSC [1]; learning based detectors, such as ConvNet [29]; and the deep models YOLOv3 [18], FRCNN [19], FRCNN + BN [19], SAR R-CNN [30], and RPN-BF [31]. The AMRs shown in Table 8 are cited from [1], except for the proposed detector, ACF, and YOLOv3. As shown in the table, the proposed detector produced the lowest AMR of the listed nondeep model-based detectors. It even outperformed YOLOv3 and achieved a performance comparable to that of FRCNN for the INRIA test dataset. Other deep-model-based detectors produced much lower AMRs by taking advantage of extracting a large number of high-level features and training many epochs on high-end GPUs.

#### 3.2.2. Evaluation of Robustness

To evaluate the robustness of the integrated detector, it was tested on the ETHZ dataset, Caltech test set, and Citypersons validation set in the following experiments. The parameters of the integrated detector were still $offset=0,r_{w}=3,r_{h}=7,\;f_{wr}=0.5,$ $f_{hr}=0.125,$ $d_{hr}=0.25$, and $f_{ss}=8$, $s_{thr}=25$. The embedded ACF detector was pretrained on the INRIA dataset or the Caltech dataset, if specified.

The results for the two sequences of the ETHZ dataset (Table 9 and Table 10) show that the proposed integrated detector produced more true positives and fewer false positives, leading to an increased recall and decreased AMR positives. For the Caltech test set, a decrease in AMRs (Table 11) was also observed.

A cross dataset evaluation on the Citypersons dataset was performed. The detectors were first pretrained on the INRIA and Caltech datasets and were then tested on Citypersons validation set. It was observed that the integrated detector significantly decreased the AMR (Figure 23a–c and Figure 24a–c) and increased the recall (Figure 23d–f and Figure 24d–f) for the reasonable, heavily occluded, and all subsets with both pretrained datasets. This, together with the information shown in Table 9 and Table 10, indicates that the proposed integrated detector as well as the parameter design are robust when used with the training set and are applicable for use on an unseen dataset.

We also investigated the robustness of the proposed detector under different thresholds. The results (Table 11, Figure 23 and Figure 24) show that the integrated detector improved the performance under varying thresholds, and the improvement was more significant under higher thresholds. This means that the proposed method is not only robust to thresholds but performs better when fewer false positives are required.

The influence of the minimum size (denoted as ms) of faces on the integrated detector was studied. The smaller the parameter ms, the more faces MTCNN was able to detect. According to Table 11 and Figure 23 and Figure 24, more noticeable improvements were observed in most cases, except for the reasonable and all subsets pretrained on INRIA when ms was decreased to 13 pixels. This means that the performance of the face detector influences the improvements of the whole integrated detector.

#### 3.2.3. Detection Speed

With regard to the detection speed, the time costs of detecting 288 INRIA pedestrian test images for different detectors via Intel(R) Core (TM) i7-8565U CPU @ 1.80GHz are compared in Table 12. In terms of the time cost required to detect the INRIA test set, the integrated detector requires approximately 74.32 s to detect 288 INRIA pedestrian test images, including 51.33 s for faces. This time cost for the MTCNN detector can be compressed to 45.24 s if the O-Net is not implemented. By removing the O-Net from the integrated detector, 3 more true positives were detected at the expense of 25 more false positives for the INRIA pedestrian test dataset. Two samples of increased true positives are shown in Figure 25. To increase the true positive rate while maintaining the least false positive rate, the complete MTCNN was utilized in our integrated detector.

## 4. Conclusions

In conclusion, we presented an integrated pedestrian detector, namely the MTCNN + ACF detector, for small pedestrian datasets. It detects human bodies considering not only color and edge information but also facial features. The integrated detector aggregates multiple detection tasks uniformly to produce a final overall score, which is the sum of the scaled face score and the body score. The fusion rules and parameter choices were investigated in depth. The idea is simple and easy to implement, but the proposed detector can effectively and robustly improve the detection performance compared to the sole use of ACF detector on various pedestrian datasets. The proposed detector only utilizes the CPU device and does not require any further training; however, it achieves a performance level (14.29%) comparable to deep models such as FRCNN (14%) and YOLOv3 (14.75%) on the small pedestrian dataset. The recall and average miss rate were observed to have a steady increase and decrease, respectively, on the Citypersons, ETHZ, and Caltech datasets. Therefore, the proposed detector is an effective paradigm of multitask collaboration, and it serves as a cost-effective choice for pedestrian detection in the case of limited data and computational resources.

## Figures and Tables

**Figure 1 sensors-22-03568-f001:**
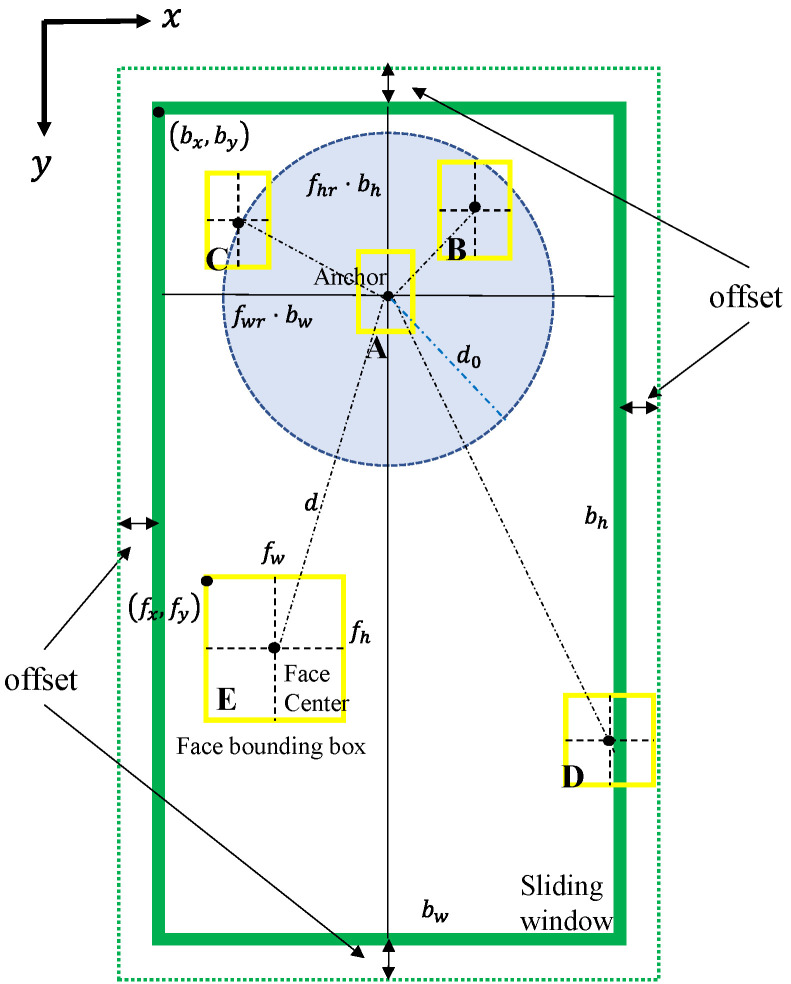
The depiction of the sliding window (green solid box), the face bounding boxes (yellow solid boxes A to E), and the parameters used for the score combination. The edge of the blue-shaded circle marks out the zero-scaled score positions. The scaled scores are positive when face centers are inside this circle and negative when the centers are outside.

**Figure 2 sensors-22-03568-f002:**
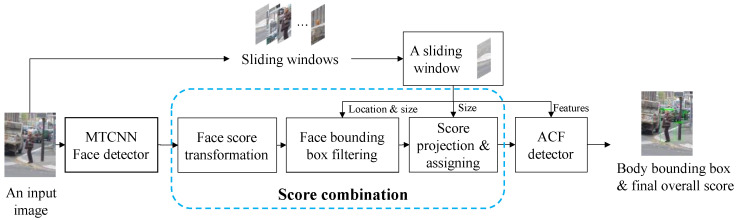
The workflow of the proposed human body detector.

**Figure 3 sensors-22-03568-f003:**
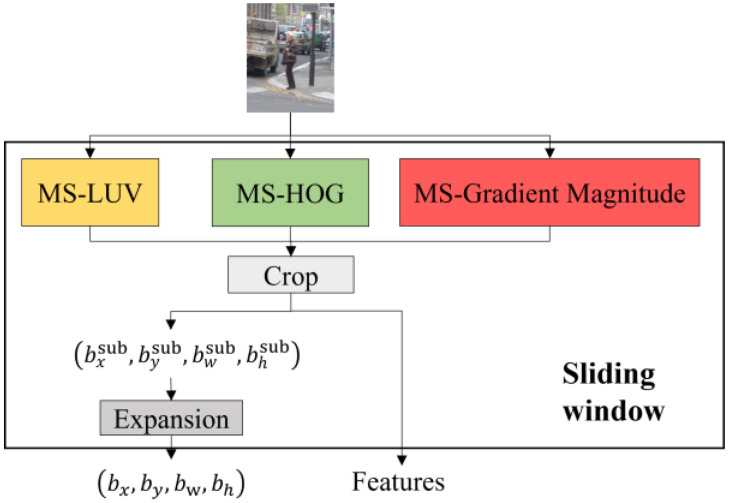
The diagram of the sliding window module. Note that ‘MS’ stands for multiscale.

**Figure 4 sensors-22-03568-f004:**
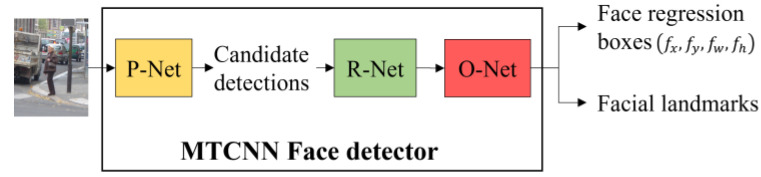
The diagram of the MTCNN face detector.

**Figure 5 sensors-22-03568-f005:**
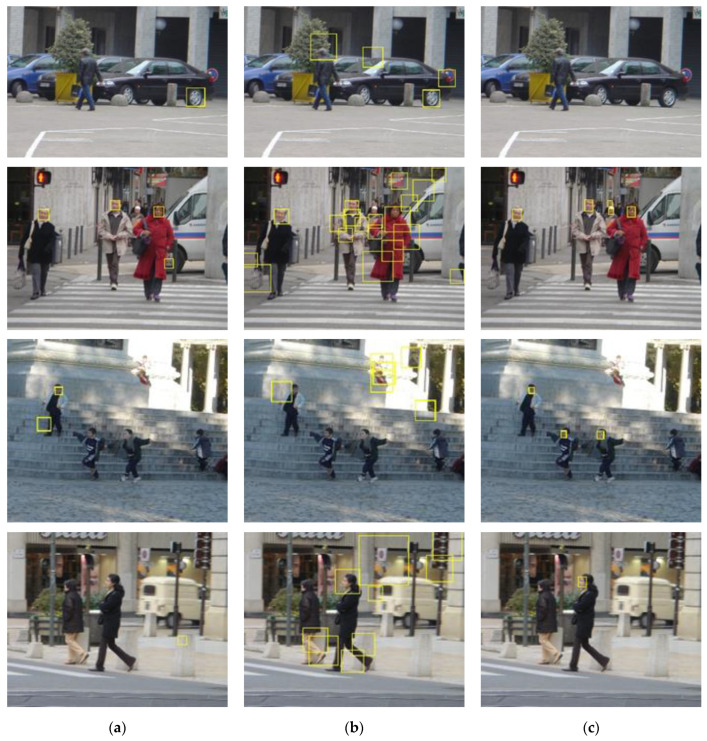
Face detection results of sample images using the VJ face detector (**a**), the fast face detector (**b**), and the MTCNN detector (**c**).

**Figure 6 sensors-22-03568-f006:**
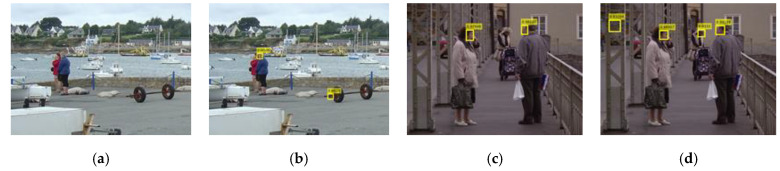
Compare to the results obtained with the complete MTCNN detector (**a**,**c**), more true faces are detected (**b**,**d**) when the O-Net is removed.

**Figure 7 sensors-22-03568-f007:**
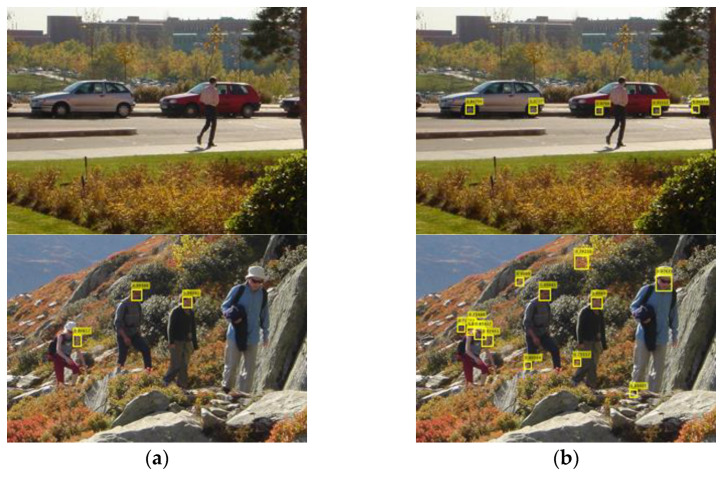
The O-Net assists with the correction of face bounding boxes by the estimation of facial landmarks. We observe that the MTCNN detector produces fewer false faces (**a**), although some faces with occluded facial features are missed. In contrast, a large number of false faces appear in the complex backgrounds (**b**) when the O-Net is removed from the MTCNN detector. We observed that these are associated with particular structures, for example, car wheels.

**Figure 8 sensors-22-03568-f008:**
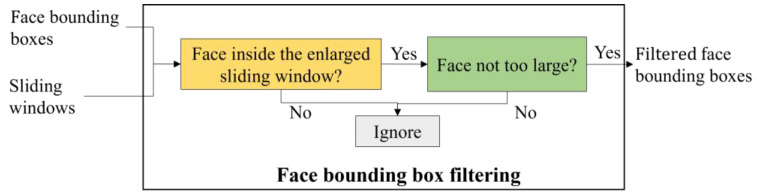
Diagram of the face bounding box filtering module.

**Figure 9 sensors-22-03568-f009:**
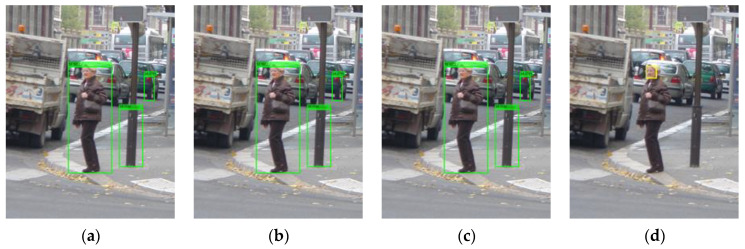
Detection results of the ACF detector (**a**), the integrated detector with offset=0 (**b**) and offset=5 (**c**), and the MTCNN detector (**d**).

**Figure 10 sensors-22-03568-f010:**
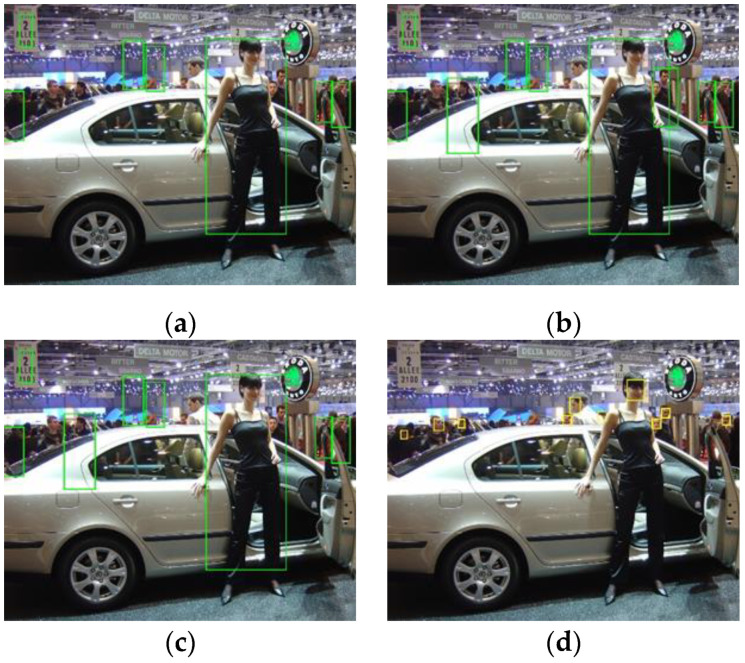
Detection results of the ACF detector (**a**), the integrated detector with $r_{w}=r_{h}=0$ (**b**) or $r_{w}=3$, $r_{h}=8$ (**c**) and the MTCNN detector (**d**). In (**b**), there are 2 more false positives than in (**a**), while (**c**) only has one more false positive.

**Figure 11 sensors-22-03568-f011:**
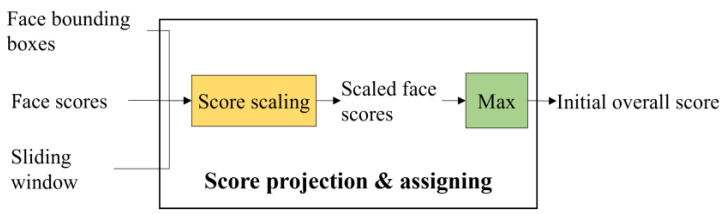
Diagram of the score projection and assigning module. Note that the filtered face information is fed into this module. The output initial overall score is assigned to the corresponding input sliding window.

**Figure 12 sensors-22-03568-f012:**
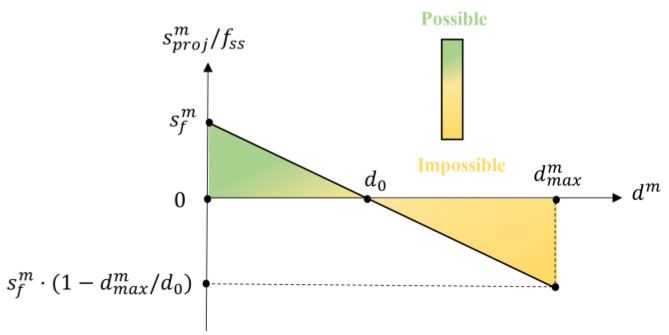
The scaled score divided by $f_{ss}$ versus the distance from the m -th face center to the anchor is drawn according to (8). The higher the scaled score is, the more likely the sliding window is to contain a human body.

**Figure 13 sensors-22-03568-f013:**
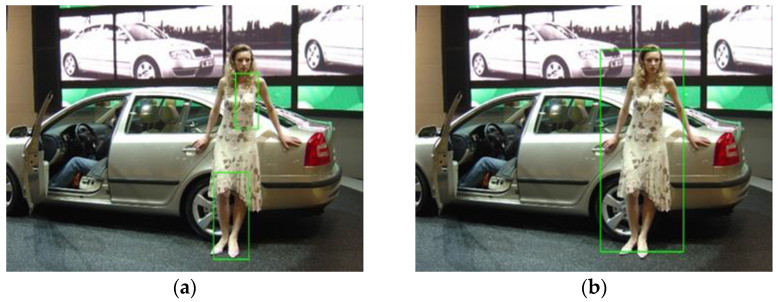
The human body is missed by the ACF detector (**a**), but it is detected by the integrated detector with $f_{ss}=1$ (**b**).

**Figure 14 sensors-22-03568-f014:**
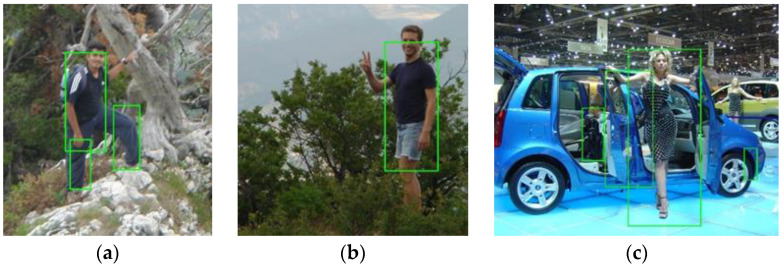
The human bodies are missed (**a**–**c**) when $f_{ss}=1$, even though their faces are correctly detected (**d**–**f**).

**Figure 15 sensors-22-03568-f015:**
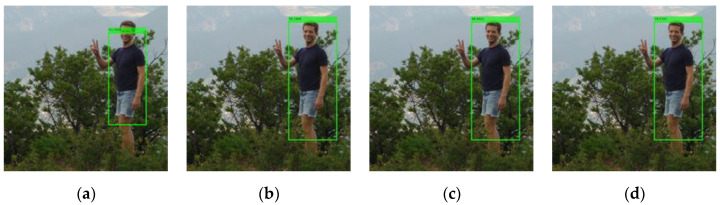
The detection results of the integrated detector with $f_{ss}=1,\;5,\;8,\;10$ for (**a**–**d**) respectively.

**Figure 16 sensors-22-03568-f016:**
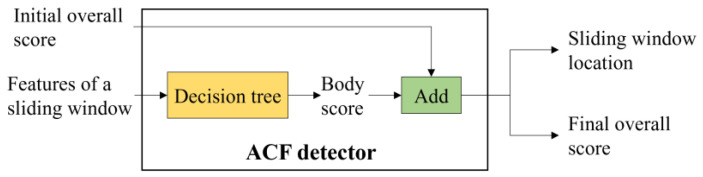
Diagram of the ACF detector.

**Figure 17 sensors-22-03568-f017:**
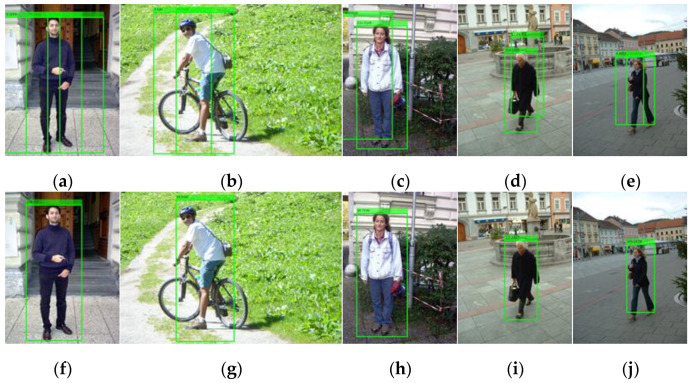
The results of the MTCNN + ACF detector with $f_{ss}=8$ (**a**–**e**) and the ACF detector (**f**–**j**). Both detectors utilize the default threshold.

**Figure 18 sensors-22-03568-f018:**
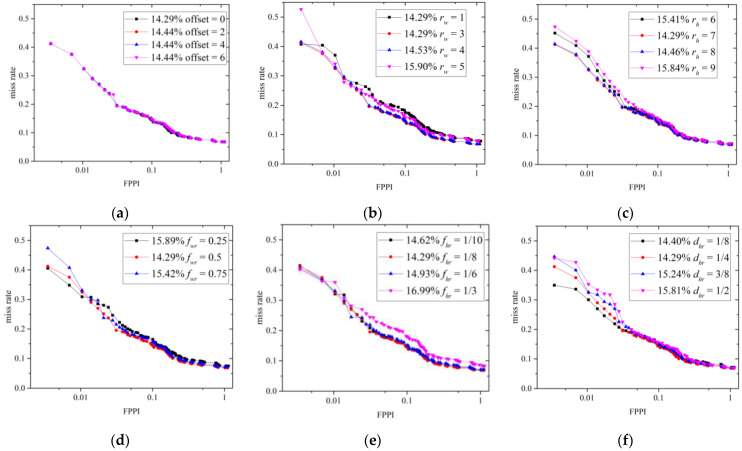
Miss rate versus FPPI for different choices of offset (**a**), $r_{w}$ (**b**), $r_{h}$ (**c**), $f_{wr}$ (**d**), $f_{hr}$ (**e**), and $d_{hr}$ (**f**).

**Figure 19 sensors-22-03568-f019:**
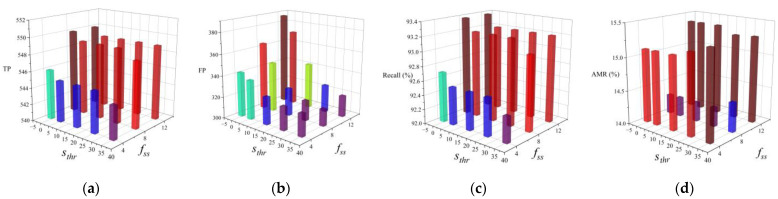
The choices of $s_{thr}$ and $f_{ss}$ mutually influence the number of true positives (**a**), the number of false positives (**b**), the recall (**c**), and the average miss rate (**d**).

**Figure 20 sensors-22-03568-f020:**
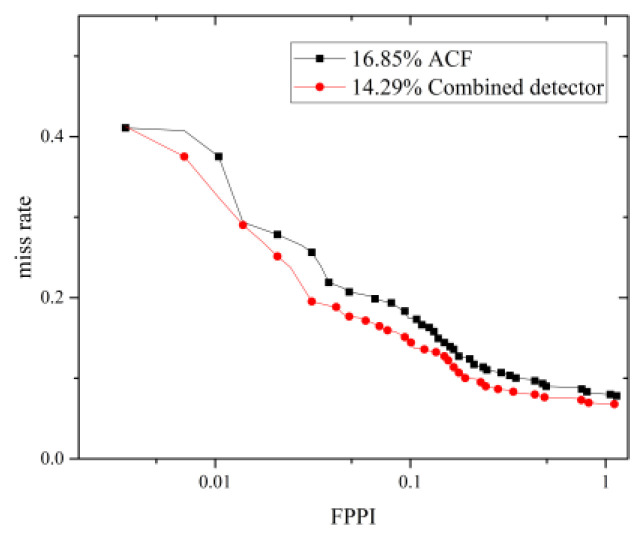
Miss rate versus FPPI for the ACF detector and the proposed integrated detector on the INRIA pedestrian test set.

**Figure 21 sensors-22-03568-f021:**
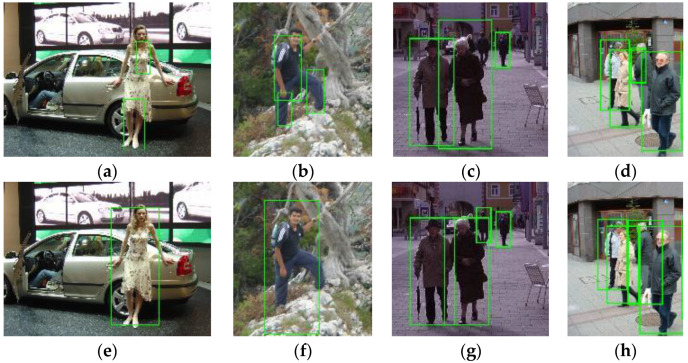
The integrated detector recognizes some human bodies (**e**–**h**) that are missed by the ACF detector (**a**–**d**).

**Figure 22 sensors-22-03568-f022:**
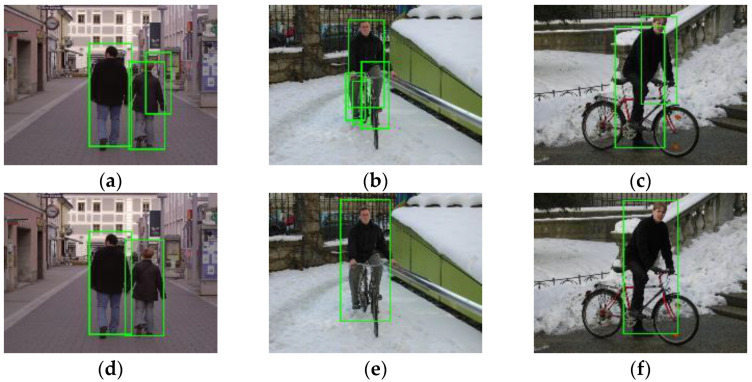
The integrated detector yields fewer false human bodies (**d**–**f**) than the ACF detector (**a**–**c**).

**Figure 23 sensors-22-03568-f023:**
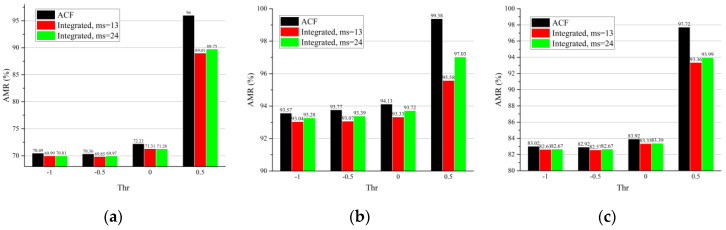
The AMR and recall of the reasonable subset (**a**,**d**), heavily occluded subset (**b**,**e**), and all subset (**c**,**f**) for the the Citypersons validation set. The ACF detector was pretrained on the Caltech dataset.

**Figure 24 sensors-22-03568-f024:**
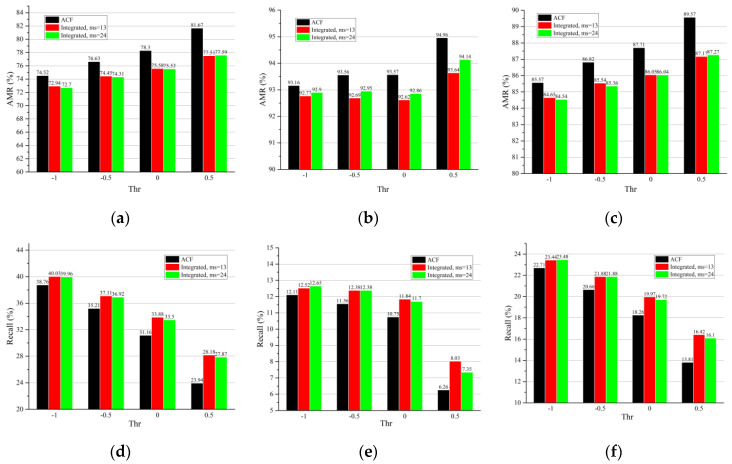
The AMR and recall of the reasonable subset (**a**,**d**), heavily occluded subset (**b**,**e**), and all subset (**c**,**f**) for the Citypersons validation set. The ACF detector was pretrained on the INRIA dataset.

**Figure 25 sensors-22-03568-f025:**
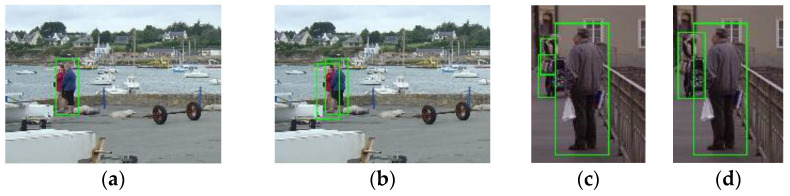
By removing the O-Net from the integrated detector, more human bodies ($f_{ss}=20$, $s_{thr}=16$ ) were detected (**b**,**d**) than with the complete integrated detector (**a**,**c**).

**Table 1 sensors-22-03568-t001:** Comparison of different offset values in the integrated detector.

offset	TP	FP	R (%)	AMR (%)
0	549	319	93.21	14.29
2	549	322	93.21	14.44
4	549	322	93.21	14.44
6	549	322	93.21	14.44

**Table 2 sensors-22-03568-t002:** Comparison of different $r_{w}$ values in the integrated detector.

$r_{w}$	TP	FP	R (%)	AMR (%)
1	549	319	93.21	14.29
3	549	319	93.21	14.29
4	549	314	93.21	14.53
5	542	308	92.02	15.90

**Table 3 sensors-22-03568-t003:** Comparison of different $r_{h}$ values in the integrated detector.

$r_{h}$	TP	FP	R (%)	AMR (%)
6	547	321	92.87	15.41
7	549	319	93.21	14.29
8	549	316	93.21	14.46
9	547	308	92.87	15.84

**Table 4 sensors-22-03568-t004:** Comparison of different $f_{wr}$ values in the integrated detector.

$f_{wr}$	TP	FP	R (%)	AMR (%)
0.25	545	328	92.53	15.89
0.5	549	319	93.21	14.29
0.75	547	317	92.87	15.42

**Table 5 sensors-22-03568-t005:** Comparison of different $f_{hr}$ values in the integrated detector.

$f_{hr}$	TP	FP	R (%)	AMR (%)
1/10	548	321	93.04	14.62
1/8	549	319	93.21	14.29
1/6	547	316	92.87	14.93
1/3	540	327	91.68	16.99

**Table 6 sensors-22-03568-t006:** Comparison of different $d_{hr}$ values in the integrated detector.

$d_{hr}$	TP	FP	R (%)	AMR (%)
1/8	547	323	92.87	14.40
1/4	549	319	93.21	14.29
3/8	548	322	93.04	15.24
1/2	548	330	93.04	15.81

**Table 7 sensors-22-03568-t007:** Comparison of the detection results of the ACF detector and the integrated detector for the INRIA pedestrian test dataset.

	TP	FP
ACF	543	328
Integrated detector	549	319

**Table 8 sensors-22-03568-t008:** Comparison of the AMRs of state-of-the-art detectors based on handcrafted features and deep models to the proposed detector for the INRIA test set.

Method	Deep-Model-Based Body Detector	AMR (%)
HOG + SVM [13]	No	45.18
DPM [16]	No	19.96
ConvNet [29]	No	17.1
ACF [14]	No	16.85
YOLOv3 [18]	Yes	14.75
ACF + HSC [1]	No	14.38
Integrated detector (proposed)	No (except for face detection)	14.29
FRCNN [19]	Yes	14
FRCNN + BN [19]	Yes	12
SAR R-CNN [30]	Yes	8.04
RPN-BF [31]	Yes	6.9

**Table 9 sensors-22-03568-t009:** Detection results for the BAHNHOF sequence.

	TP	FP	R (%)	AMR (%)
ACF	2736	1900	81.89	48.79
Integrated detector	2743	1867	82.10	46.04

**Table 10 sensors-22-03568-t010:** Detection results for the Sunny Day sequence.

	TP	FP	R (%)	AMR (%)
ACF	1250	91	80.13	31.90
Integrated detector	1268	84	81.28	28.82

**Table 11 sensors-22-03568-t011:** AMRs (%) of the reasonable subset for the Caltech test set. The ACF detector was pretrained with the Caltech training set.

Thr ^1^	−1	−0.5	0	0.5
ACF	24.41	24.41	27.77	89.01
Integrated detector (ms ^2^ = 24)	24.39	24.14	27.77	88.89

^1^ The threshold of the decision forest. ^2^ The minimum size of faces that the MTCNN can detect.

**Table 12 sensors-22-03568-t012:** The time cost required to detect the INRIA test set.

Method	Proposed	Proposed ^1^	DPM	ACF	HOG + SVM	YOLOv3
Time cost (s)	22.99 + 51.33	22.99 + 45.24	505.26	17.13	45.98	467.41

^1^ Proposed detector without O-Net.

## Data Availability

Not applicable.

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
