# Peer review of "Pedestrian Detection Using Integrated Aggregate Channel Features and Multitask Cascaded Convolutional Neural-Network-Based Face Detectors"

_sensors, 2022, doi:10.3390/s22093568_

Round 1

Reviewer 1 Report

The work is interesting and inspiring to the field of pedestrian detection. The author gives a large number of and sufficient experimental verification. I therefore recommend this paper to be published. And it is better if the authors consider the following mentioned remarks and further improve the manuscript before submitting the final version. My detailed comments are as follows:

1. Section 3.2.3, the content of detection speed is relatively simple. It would be better if you could add speed comparison with other similar detectors.

2. Figure 2 shows the overall structure of the integrated detector. The detailed structure diagram of each component can be added in the detailed description of  Section 3.

3. The description of "Introduction" needs to be improved. In addition to introducing the two integrated detectors, the detailed work the author had done to integrate the two should also be highlighted, for example: "Score combination"

Reviewer 2 Report

Abstract should denote the impact of the work to demonstrate it's importance.

Good use of language presented. Overall very clear on what has been presented.

However, language use can be colloquial at points, such as line 142: "heavily occluded" should be "significantly occluded" or similar. Please review and correct throughout as appropriate.

Literature review sets the scene and presents a valid current state of the problem and current techniques being undertaken.

Abbreviations are declared inconsistently with mixed letter case: Type Case and lower case are used. Correct throughout to be consistent. An example in line 90-91 where two abbreviations are declared with inconsistent format.

Equation 8 should be defined with mathematic notation to describe recall, otherwise explain in text as the equation is already described in a verbose English word manner and includes a full stop. 

Detailed exploration of results undertaken with an adequate narrative.

Table 8 should include references to support comparison with other current work.

Conclusion should summarise some results to support narrative and demonstrate impact.
